# Deep Learning-Based Segmentation of Post-Mortem Human's Olfactory Bulb Structures in X-ray Phase-Contrast Tomography

Alexandr Meshkov [1], Anvar Khafizov [2,3], Alexey Buzmakov [2,4], Inna Bukreeva [5,6], Olga Junemann [7], Michela Fratini [5,8], Alessia Cedola [5], Marina Chukalina [2,9,10,*], Andrei Yamaev [9], Giuseppe Gigli [11], Fabian Wilde [12], Elena Longo [13], Victor Asadchikov [2], Sergey Saveliev [7] and Dmitry Nikolaev [9,10]

1 The Moscow Institute of Physics and Technology, 9 Institutskiy per., 141701 Moscow, Russia; meshkov.a@phystech.edu

2 FSRC «Crystallography and Photonics» RAS, Leninskiy pr. 59, 119333 Moscow, Russia; ankhafizov1998@yandex.ru (A.K.);buzmakov@gmail.com (A.B.); asad@crys.ras.ru (V.A.)

3 Croc Inc. Company, Volochayevskaya Ulitsa 5/3, 111033 Moscow, Russia

4 Federal Research Center "Computer Science and Control" of the Russian Academy of Sciences, Vavilova Str. 44b2, 119333 Moscow, Russia

5 Institute of Nanotechnology—CNR, c/o Department of Physics, La Sapienza University, Piazzale Aldo Moro 5, 00185 Rome, Italy; inna.bukreeva@cnr.it (I.B.); michela.fratini@cnr.it (M.F.); alessia.cedola@cnr.it (A.C.)

6 P.N. Lebedev Physical Institute, RAS, Leninskiy pr. 53, 119991 Moscow, Russia

7 FSSI Research Institute of Human Morphology, Tsyurupy Str. 3, 117418 Moscow, Russia; o.junemann@mail.ru (O.J.); braincase@yandex.ru (S.S.)

8 IRCCS Santa Lucia Foundation, Via Ardeatina 306/354, 00142 Rome, Italy

9 Smart Engines Service LLC, 60-Letiya Oktyabrya pr. 9, 117312 Moscow, Russia; a.yamaev@smartengines.com (A.Y.); dimonstr@iitp.ru (D.N.)

10 Institute for Information Transmission Problems of Russian Academy of Sciences (Kharkevich Institute), Bol'shoi Karetnii per. 19 Str. 1, 127051 Moscow, Russia

11 Institute of Nanotechnology—CNR, c/o Campus Ecotekne—Universita del Salento, Via Monteroni, 73100 Lecce, Italy; giuseppe.gigli@cnr.it

12 Institute of Materials Research, Helmholtz-Zentrum Hereon, Max-Planck-Str. 1, 21502 Geesthacht, Germany; fabian.wilde@hereon.de

13 Elettra-Sincrotrone Trieste S.C.p.A., 34149 Trieste, Italy; elena.longo@elettra.eu

* Correspondence: chukalinamarina@gmail.com

**Abstract:** The human olfactory bulb (OB) has a laminar structure. The segregation of cell populations in the OB image poses a significant challenge because of indistinct boundaries of the layers. Standard 3D visualization tools usually have a low resolution and cannot provide the high accuracy required for morphometric analysis. X-ray phase contrast tomography (XPCT) offers sufficient resolution and contrast to identify single cells in large volumes of the brain. The numerous microanatomical structures detectable in XPCT image of the OB, however, greatly complicate the manual delineation of OB neuronal cell layers. To address the challenging problem of fully automated segmentation of XPCT images of human OB morphological layers, we propose a new pipeline for tomographic data processing. Convolutional neural networks (CNN) were used to segment XPCT image of native unstained human OB. Virtual segmentation of the whole OB and an accurate delineation of each layer in a healthy non-demented OB is mandatory as the first step for assessing OB morphological changes in smell impairment research. In this framework, we proposed an effective tool that could help to shed light on OB layer-specific degeneration in patients with olfactory disorder.

**Keywords:** olfactory bulb; deep learning; convolutional neural network; segmentation; X-ray phase-contrast tomography

## 1. Introduction

The mammalian olfactory bulb (OB) is a part of the forebrain involved in olfaction. It plays an essential role in the sense of smell. OB is the first control centre in the olfactory

path for processing and for relaying odour information to the piriform (olfactory) cortex. The sense of smell projects directly to the limbic system, a part of the brain responsible for emotion, behaviour, long-term memory, and olfaction.

In recent decades, olfaction in humans and animals has been extensively studied via a wide variety of approaches involving behavioural symptom analysis, histopathology, the study of structural and molecular alterations in the olfactory system, as well as the investigation of genetic and environmental influences on olfaction [1–3]. However, the study of morphological changes in the human OB accompanying olfactory dysfunction is still a hotly-debated topic [4,5].

OB is arranged into several nested layers, each with a specific neural structure. Accurate image segmentation of the normal OB and correct delineation of each of its microanatomical structures, is mandatory as the first step for assessing OB morphological changes in smell impairment research.

Due to of indistinct layer boundaries, differentiation of cell populations in the OB image is challenging issue. Consequently, morphometric data analysis using conventional low-resolution 3D imaging tools [6] and manual delineation of OB areas are difficult to perform. A number of recent studies have demonstrated that micro-CT is able to provide high-quality images of soft tissue morphology [7]. However, this imaging technique, which is effective when using staining agents, does not allow soft tissue differentiation based on native tissue contrast.

We applied advanced X-ray phase contrast tomography (XPCT) as a reliable non-destructive 3D imaging tool to access morphological structures of uncut and unstained post-mortem human OB. Due to high sensitivity to weak perturbation of the X-ray wavefront, XPCT enhances the visibility of inner details in unstained biological soft tissues. A great benefit of XPCT is the ability of high-resolution 3D imaging of the whole unstained OB down to the cell and capillary level. However, numerous microanatomical structures in OB such as neural cell bodies, their processes, blood vessels, etc., create operational challenges for the manual delineation of macro-scale OB features such as OB neuronal cell layers. In addition, segmentation slice by slice [8–10] could be very labor-intensive due to the large amount of data involved.

We present the pipeline for fully automatic segmentation of neural cell type-specific layers in XPCT imaging of post-mortem human OB. The developed method is based on a Deep learning (DL) model. Deep learning is a class of efficient computational algorithms using a multi-layer artificial neural network. Recently, DL has emerged as an effective tool in data-rich research areas such as biomedical tomographic imaging [1]. In particular, it is a highly promising method for the segmentation, identification, and quantification of morphological structures in the brain [11,12].

Convolutional neural network (CNN) is a class of deep neural network typically used to address challenges in computer vision applications [13–21]. In particular, different architectures of CNN have been efficiently used in medical and biomedical brain images segmentation [22–24]. We used CNN as the core for semantic segmentation of morphological features in human OB. This work represents a proof of concept for the development of a segmentation tool for high resolution XPCT images. We collected training and validation datasets from XPCT images of the OB. To ensure stable results, we used a two-step segmentation approach. A U-net architecture [25] with a categorical cross-entropy loss function [26] has been adapted to build the models for segmentation. The first model is to perform two-class segmentation (to extract OB from background). The second model is to perform 5-classes segmentation (to extract the morphological layers within the OB). A categorical cross-entropy loss function was selected, as it is commonly used and has good convergence with gradient descent. We trained the U-Net models with prepared labelled data. The quality of the performance with Dice coefficient [27] (as a statistical validation metric) was 0.98 for two-class segmentation and 0.84 for five-classes segmentation, respectively. We built and trained CNN models to be a useful computational tool for fully automatic layer-specific segmentation of human OB XPCT images.

DL-based methods have largely been used for computer-aided detection in various pathologies. As our model is capable of processing high-resolution images of soft tissue, we believe it to be an important research tool for the advancement of clinical imaging techniques. This can also be extended to identify biomarkers related to neurodegenerative diseases in the future.

This paper is organized as follows. Section 2 includes the next parts. First, we described the sample and preparation procedures for XPCT and immunohistochemistry. Next, the experimental setup for obtaining XPCT projections as well as 3D image reconstruction method were detailed. The statement of segmentation problem and artificial neural network approach description complete the section. Section 3, describes a pipeline aiming to segment morphological structures of human OB in the XPCT image. Here, we demonstrate high resolution XPCT and immunohistochemical images of the OB and provided details of neural network model building for automatic segmentation of the OB images: train and test data preparation, CNN training process, and the results of the OB slices segmentation are described. We discuss the application of obtained results, and consider possible future studies of the OB structure in Section 4. We compare CNN segmentation with traditional thresholding segmentation techniques in the Supplementary Materials.

## 2. Materials and Methods

### 2.1. Pipeline for Semantic Segmentation of Morphological Structure in Human OB

We have developed and used a pipeline for semantic segmentation of morphological structure in human OB presented in Figure 1. An important part of the pipeline is a building of CNN model for automatic segmentation of the OB layers, including XPCT slices annotation, ground estimation, and model training. The pre-trained model was applied for fully automated segmentation of the OB morphological layers.

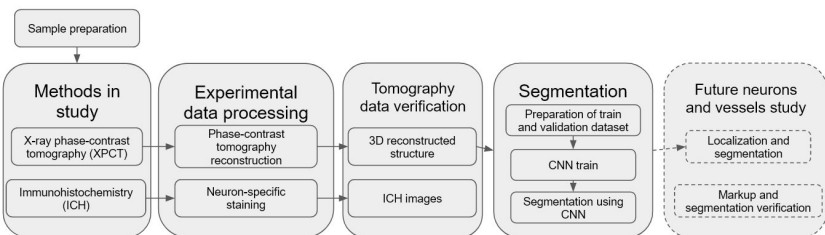

**Figure 1.** Illustration of the pipeline for segmentation of the OB using CNN.

### 2.2. Sample Description

OB tissue sample was taken from the post-mortem human brain of a non-demented elderly person.

### 2.3. Sample Preparation for XPCT and Immunohistochemistry

The sample was fixed in formalin, then was dehydrated in 8 portions isopropyl-alcohol and embedded in cylindrical paraffin blocks $5 \times 5 \times 10$ mm$^3$ dimension, as for routine histological examination, and the block was measured via XPCT. After XPCT experiments, the samples were prepared for immunohistochemical analysis. Paraffin blocks were cut at 6-mm thickness sections. The sections were incubated with primary antibodies to $\beta$-III-tubulin (dilution 1:500, Thermo Fisher Scientific, Waltham, MA, USA) and PGP9.5 (dilution 1:300, Thermo Fisher Scientific). The UltraVision Quanto Detection System kit by Thermo Fisher Scientific was used as the detection system. These sections were examined microscopically to identify areas within the blocks that contained histological features of interest – namely, neurons of different types forming layers of the OB. The Zeiss A1, made in Germany, which is a light compound microscope, was used for the microscopic measurements.

### 2.4. X-ray Phase-Contrast Tomography Experimental Setup

We used a propagation-based XPCT setup to image a human post-mortem OB in 3D. XPCT relies on the phase shift of the X-ray beam as it passes through a sample. Although different image contrast-enhancing techniques proved to be efficient tools in visualizing the biological sample, the propagation-based XPCT has been demonstrated to be highly efficient in 3D virtual histology on human and animal cerebellar tissue [28,29].

The XPCT experiment was carried out at the P05 beamline of the synchrotron facility PETRA III, DESY operated by the Helmholtz-Zentrum Hereon [30,31]. An X-ray energy of about 25 keV was selected with the Double Crystal Monochromator (Si111). A step-wise tomographic scan was performed with 4000 projections equispaced within a 360 degree range. The exposure time was about 0.07 s. Half-acquisition extended mode was used to increase the tomography field of view. The sample-to-detector distance was set at 50 cm. Projection images were detected with magnification , resulting in an effective pixel size $1.28 \times 1.28$ μm$^2$. We applied in the recostructed data binning $2 \times 2$ to increase signal to noise ratio (binned pixel size about of $2.6 \times 2.6$ μm$^2$)

### 2.5. Tomography Reconstruction

The raw X-ray projection data were pre-processed with the flat- and dark-fields correction, and a phase-contrast retrieval was applied to each projection:

$$T = \frac{2}{z\lambda} \left( F^{-1} \left\{ \frac{F[I(x,y,z=D)/I_0 - 1]}{\left( k_x^2 + k_y^2 + \alpha \right)} \right\} \right) \tag{1}$$

where $\alpha$ is a parameter that regulates the trade-off between the blurring and suppression of phase-contrast-induced fringes, $\lambda$ is the wavelength corresponding to the energy E. $I_0$ is the incident beam intensity; however, for practical reasons, the intensity was measured at distance $D$ without the sample, and this is thus just an approximation of $I_0$ [32]. As a result of Equation (1), we used a linearization of the transport-of-intensity equation with a regularization at zero frequency. Tomographic reconstruction was conducted with the Filtered Back Projection method (FBP) implemented in the tomographic reconstruction pipeline. The size of the reconstructed tomographic slice, accounting for binning $2 \times 2$, was $1460 \times 1460$ pixels. In the CNN model, the input image size was further reduced to 50%. As a result, the images used in our model were $730 \times 730$ pixels (about $3.7 \times 3.7$ mm).

### 2.6. Segmentation

Semantic segmentation [33] is generally defined as the classification of each pixel in an image into one of predefined several classes based on specific criteria. In order to segment OB layers in XPCT images, we use a CNN-based approach. A two-step segmentation algorithm was approved to provide a more stable solution:

1.  OB image is segmented from background elements, such as paraffin and air. In this step, the XPCT image is segmented into two classes: objects (OB) and background (paraffin and air).
2.  We performed multiclass segmentation with 5 classes, one for each OB layer to be segmented. Each pixel of the OB image is labelled with a corresponding class (OB layer).

In this framework, to create ground truth data, biologists manually annotated the areas of each layer in 300 slices from the volumetric OB grayscale reconstructed image (4000 slices). Then, the annotated dataset was divided into training and validation parts. Training and testing datasets were prepared using SurVoS software [34], which provides the option to load reconstructed XPCT slices, annotate regions of interest (ROI), and convert them to binary or multiclass masks for training a CNN model.

*2.7. CNN Architecture*

Basically, CNN is a series of linear and non-linear mathematical operations such as convolutions, pooling, etc., with learnable parameters (weights) in each layer of the network. The parameters are not predetermined and depend on the problem to be solved. If CNN is provided with the correct weights, it will generate the correct images. Otherwise, there will be an erroneous result. Therefore, the search of correct CNN weight values is the primary concern in establishing of our image processing procedure. The weight values in the CNN were found as follows. The training dataset was used as an input to train the CNN. The network weights, in the beginning, were randomly determined and then they were changed by minimizing a specific penalty (loss) function.

The U-Net architecture [25] comprises two parts-branches: encoder and decoder. The encoder is responsible for extracting the relevant details from the input image by "compressing" it in the one-dimensional feature vector, when the decoder constructs the segmentation mask based on the information hidden in this vector. Furthermore, each part consists of convolutions and non-linear mathematical operations (e.g., ReLU [35]). The output result of each operation on the input image is called an activation map.

The key reason why 2D U-Net was chosen is its high robustness, particularly in the case of medical imaging. Going deep into details, the stability of the segmentation is caused by a relatively small number of tuning CNN weights and the existence of skip-connections, e.g., informational channels between the activation map in encoder and decoder. Both of these features help U-Net to not overfit training data, e.g., avoid performing well on only the training dataset, while failing on the validation dataset at the same time. This is important especially in small training datasets where this issue is most critical.

The categorical cross-entropy loss function [26] was selected for the CNN training, as it has good convergence with gradient descent. In the case of binary classification, the categorical cross-entropy function approximates a binary cross-entropy function, preserving all the advantages of categorical cross-entropy. We also considered the Dice coefficient [18] as a metric to evaluate segmentation accuracy:

$$\text{Dice} = \frac{TP + TP}{TP + TP + FP + TN} = \frac{2a}{a + b} \tag{2}$$

where $TP$ is true positive classification, $FP$ is false positive, $TN$ is true negative. The value of 1 of this metric indicates the ideally correct segmentation compared to ground truth, whereas the value of 0, instead refers to the incorrect one.

We used the Root Mean Squared Propagation (RMSProp) algorithm [36], with a dynamic decrease in the learning rate, to optimize the segmentation algorithm. The momentum parameter for the optimizer was set to 0.9. The learning rate was reduced by a factor of 10 each time, when a metric has stopped improving, or loss stagnates/plateaus over two learning epochs. The improvement means that the maximum absolute value of the loss function divergence during the considered iteration interval was above the threshold set at $10^{-4}$. In order to make the model more resistant to overfit during the stochastic descent, Ridge Regression (L2 Regularization) with the parameter "weight decay" set to $10^{-8}$ was used. The following hyper-parameters were chosen to optimize U-Net. The batch size and number of epochs for binary segmentation are selected as 1 and 20, respectively. The batch size and number of epochs for multiclass segmentation are selected as 1 and 80, respectively. The source code of developed CNN and trained models is available at [37].

The models were trained on the PC with Intel Core i5-7300 HQ PC, RAM: 24 GB, Nvidia GeForce GTX 1060.

## 3. Results

*3.1. XPCT and Immunohistochemical Images*

Figure 2a illustrates the human brain with OBs (bilateral reddish structures) located at the inferior side of the cerebral hemispheres, close to the front of the brain. Histologically, OB is composed of multiple layers. Immunohistochemical image of the whole-OB coronal

sections with OB layers such as the olfactory nerve layer (ONL), glomerular layer (GL), external plexiform layer (EPL), mitral cell layer (MCL), internal plexiform layer (IPL), and granule cell layer (GL) is presented in Figure 2b. The OB layers are outlined with a red line in the figure.

Grayscale XPCT images of the tomographic slices through the reconstructed volumes of human OB are shown in Figure 2c (coronal section) and Figure 2d (sagittal section). The borders of the OB layers are outlined in red. Phase images show enhanced image contrast of the OB soft tissue, which allows revealing OB anatomic structures and organization of the cells in the OB layers.

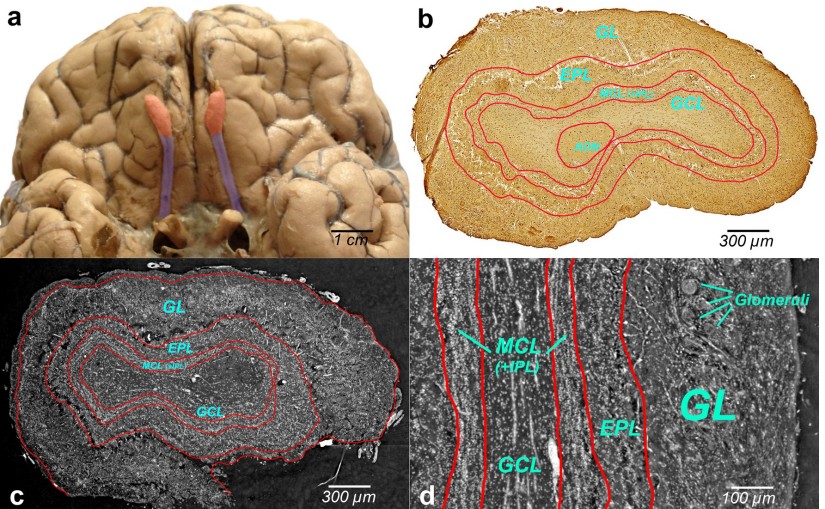

**Figure 2.** (**a**) Photographic image of the human brain with two olfactory bulbs (bilateral reddish-colored structures) and olfactory tract (violet-colored bilateral bundle of nerve fibers); (**b**) immuno-histochemical staining with antibodies to neuron-specific β-III-tubulin in human OB section (axial plane) with multi-layered cellular architecture: glomerular layer (GL), external plexiform layer (EPL), mitral cell layer & internal plexiform layer (MCL (+IPL)), granule cell layer (GL), anterior olfactory nucleus (AON); (**c**,**d**) XPCT grayscale image of the OB slice, (**c**) axial plane, (**d**) sagittal plane.

The cells within the OB have traditionally been classified according to the layers in which they reside. XPCT and immunohistochemical images of the glomeruli are shown in Figure 3a and Figure 3b, respectively.

The glomeruli (spherical structure outlined with blue lines) located in a layer just below the bulb surface (glomerular layer) is considered a first fundamental step in olfactory discrimination. These are accumulations of neuropil 100–200 μm in diameter where the axons of the olfactory nerve, supplying odour information, contact the dendrites of mitral cell (MC), periglomerular and tufted cells, the majority of these cells project an axon outside the OB to the piriform (olfactory) cortex.

Figure 3c,d show the XPCT and immunohistochemical images of the OB sagittal sections, respectively. Looking from left to right, one distinguishes the glomerular layer (GL) with multiple glomeruli and interneuron population of periglomerular cells (white cells next to the glomeruli), external plexiform layer (EPL) containing astrocytes, interneurons, some mitral cells, the mitral cell layer & internal plexiform layer (MCL (+IPL)) with mitral cell population (white cells in the laminar), and granule cell layer (GL) occupied mostly by granule cells (not shown here); the most common OB interneuron forms the OB output to other brain areas.

An important advantage of XPCT is the ability to visualize the whole OB in 3D with high resolution enabling one to see cells and capillaries. However, as illustrated in Figures 2 and 3, the complex microanatomy of the OB soft tissue poses significant difficulties in the accurate delineation of macro-scale morphological OB features such as OB layers.

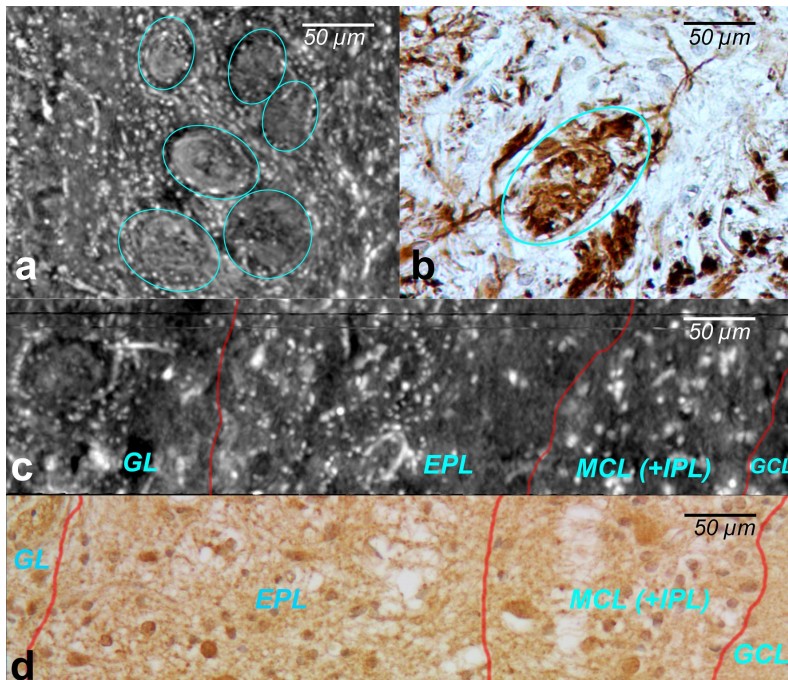

**Figure 3.** (**a**) XPCT image of the OB soft tissue (coronal section)—the glomeruli layer of the olfactory bulb. The glomeruli are the spherical structures outlined with blue lines: (**b**) immunohistochemical image of the OB soft tissue with a glomerular corresponding to (**a**); (**c**) XPCT image of the OB (sagittal section), and (**d**) immunohistochemical image corresponding to (**c**). From left to right: (GL) glomeruli in the glomerular layer, (EPL) tufted cells in the external plexiform layer, and (ML) mitral cells in the mitral cell layer.

### 3.2. Network Model/Train and Test

To differentiate morphological layers in the XPCT grayscale image of the OB, we performed a two-step CNN-based segmentation.

The first step was a foreground-background segmentation in tomography slice images. Following the expert's annotation, we generated a binary mask that can separate the foreground (OB) and background of images (paraffin and area). The second step was the segmentation of morphological layers in OB. Following the expert's annotation, we generated multiclass ROI masks, with five classes, one for each OB layer (see Section 2).

In both steps, we selected 300 tomographic slices for manual annotation resulting in datasets with pairs of OB grayscale images & annotated masks. The example of a grayscale OB image and mask manually annotated in the first step are shown in Figure 4a,b.

The prepared dataset, consisting of the original image–mask pairs, was randomly divided into training, validation, and test subsets (240, 30, and 30 pairs, respectively). The training subset was augmented, which made our network more robust. In particular, the data augmentation technique included random mirroring, random rotation within 90 degrees in the opposite directions, and adding the Gaussian noise with zero mean value and standard deviation of $10^{-4}$.

In the CNN model, each input image was resized to 50% of its original size to ease the calculations and tackle the graphic card memory overflow problem. The dataset (grayscale images paired with their masks) was used to train the CNN model and to generate binary pixel classifications. After the model training, we processed with it all slices of the volumetric OB grayscale image (4000 slices) and produced a mask for each grayscale slice. The representative image of the CNN-generated binary mask is shown in Figure 4c. The foreground (OB) is white, and the background is black. During training, segmentation accuracy was evaluated on the validation set. Figure 4d shows a grayscale tomographic slice image with OB outlined in red with a ground-truth mask (manual

segmentation) and blue with a CNN generated mask (the U-Net model). A good agreement between these two segmentations results is evident.

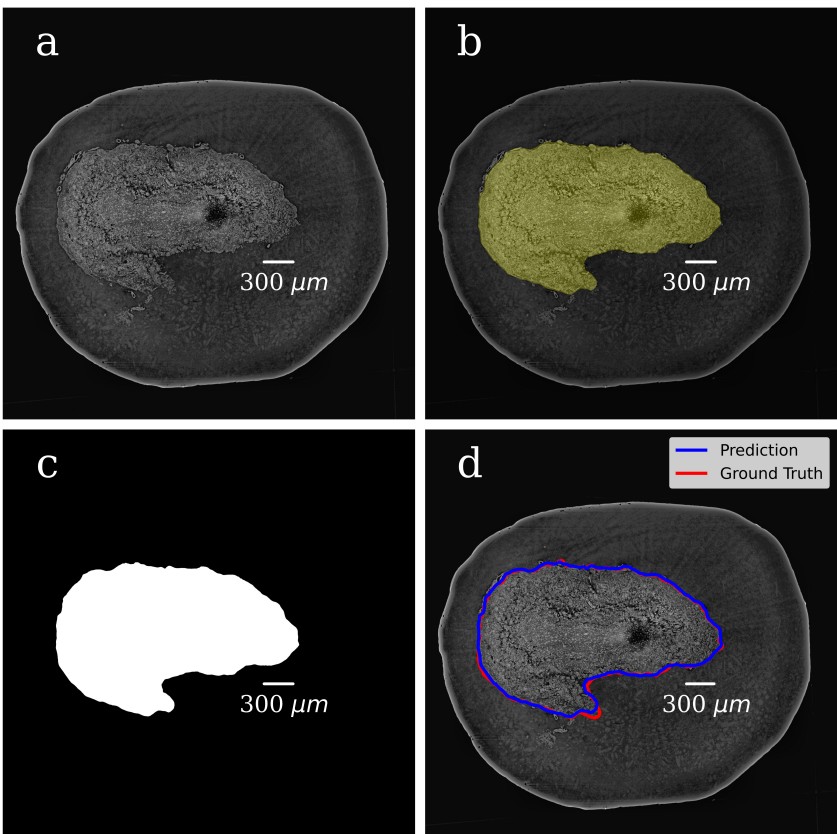

**Figure 4.** Foreground–background segmentation. (**a**) Grayscale image of the OB tomographic slice; (**b**) foreground/background mask manually annotated in the first step of segmentation, foreground (OB) is marked with greenish color, the background is the rest of the image (paraffin and area); (**c**) CNN generated mask (the U-Net model). The foreground is white, and the background is black; (**d**) grayscale slice image with OB is outlined in red with ground-truth mask (manual segmentation) and in blue with CNN generated mask (the U-Net model).

Multiclass segmentation of the OB anatomical structure (see above the 2nd step) included an additional image processing procedure. At the end of the first step, the CNN-generated binary masks were applied to correspondent grayscale images of the same size, resulting in an OB grayscale masked image where the background value was set to zero (see Figure 5a). To adapt XPCT slices thickness to histological sections, we assembled a stack of 10 neighbour tomography slices along the tomographic rotational axis (z-axis):

$$S = \{A_1, A_2, \ldots, A_{10}\}, \tag{3}$$

and for each pixel we calculated the maximum value:

$$S_{max} = max_z(\{A_1, A_2, \ldots, A_{10}\}) \tag{4}$$

As observed, $S_{max}$ could be considered as a common image of a tomographic slice because:

$$dim(S_{max}) = dim(A_1) = dim(A_2) = \ldots = dim(A_{10}) \tag{5}$$

On the other hand, the slice thickness of XPCT image Equation (4) is comparable with the histological section thickness (about 10 microns), enabling biologists to annotate training and test set images more accurately.

Based on the above, the annotation procedure included the following steps:

1. A stack of thick XPCT slices was found by stacking every 10-consecutive XPCT slices, as shown in Equation (4).
2. Training and validation subsets of thick XPCT slices was manual annotated.
3. The unstacking procedure was performed. It includes extrapolation of masks obtained with step 2 to corresponding 10-consecutive ground-truth images from step 1.

As shown by the visual control, the approach described above improved the precision of manual annotation by experts of ground-truth slices because this approach made the layer boundaries and the structure of layers themselves more evident to the annotation.

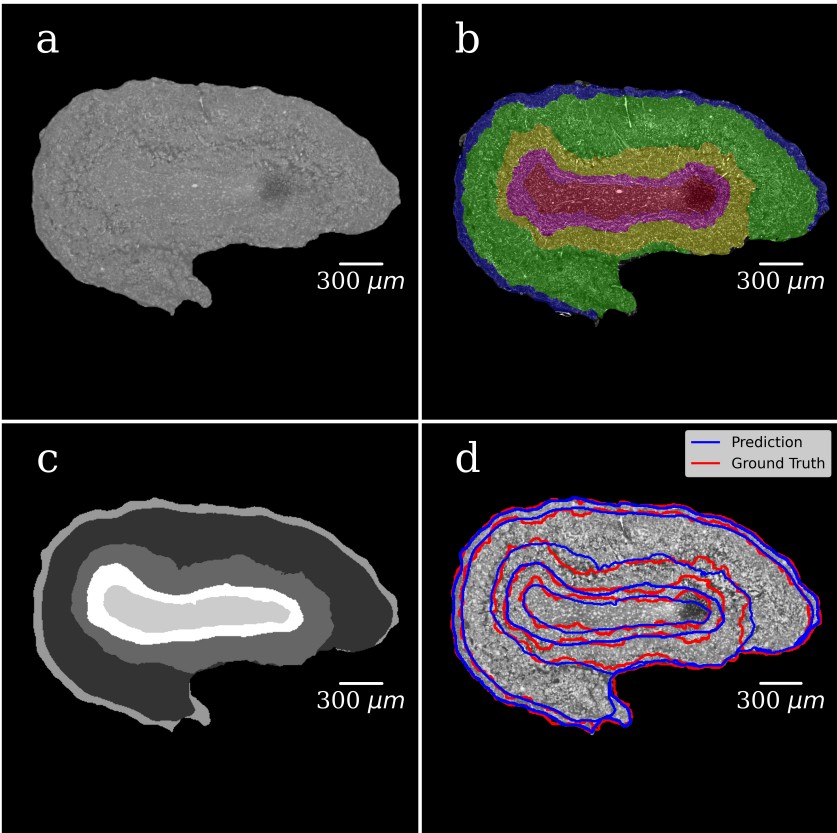

**Figure 5.** Multi-Class Segmentation. (**a**) Grayscale image of the OB tomographic slice; (**b**) multiclass mask manually annotated to segment OB anatomical layers, each color corresponds to one OB layer with a specific neural structure; (**c**) CNN generated multiclass ROI masks (the U-Net model). The grayscale of each mask corresponds to a specific OB layer; (**d**) grayscale slice image of the OB with the boundary of the OB layers outlined in red with ground-truth mask (manual segmentation) and in blue with CNN generated mask (the U-Net model).

Figure 5a shows the grayscale image of the OB segmented from the background in the first step of the segmentation. The background is set to zero. Figure 5b illustrates the manually annotated multiclass mask with five classes, one for each OB layer. Figure 5c presents CNN-generated multiclass ROI masks (the U-Net model). The gray level of each mask corresponds to a specific OB layer. Figure 5d shows an XPCT image of the OB with the boundaries between the OB layers outlined in red lines with a ground-truth mask (manual segmentation) and in blue ones with a CNN generated mask (the U-Net model). The good agreement is notable. Figure 6 shows a 3D visualisation of OB with a segmented tomographic slice.

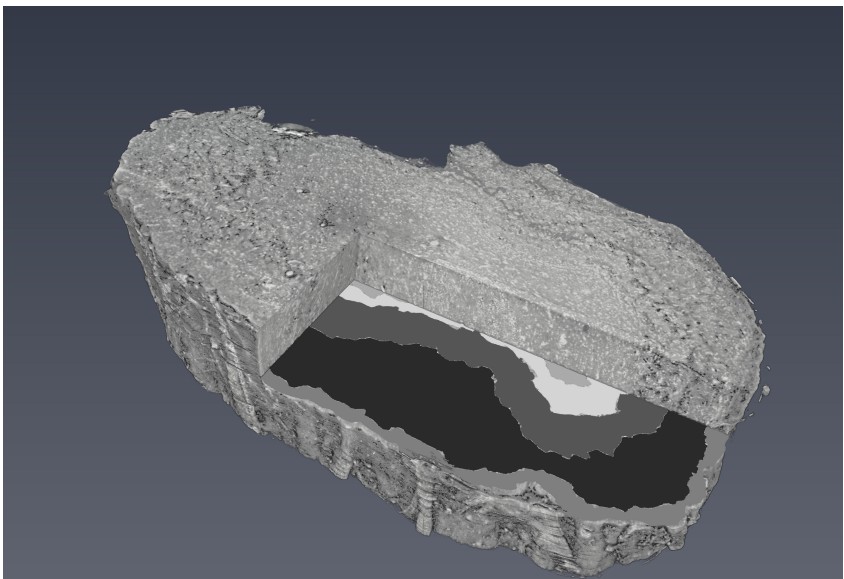

**Figure 6.** 3D visualisation of OB with segmented tomographic slice.

### 3.3. CNN training Results

To evaluate the accuracy of the grayscale image binarization performed by CNN, the Dice metric was used. Figure 7 shows the cross-validation results. In Figure 7a, the values of the Dice metric on the validation set tended to 1 starting from epoch 9. Corresponding categorical loss curves are presented in Figure 7b. It is evident from Figure 7 that there was practically no overfitting, as a high value of the Dice metric is observed on both data sets. Cross-validation Dice value for the testing is 0.98.

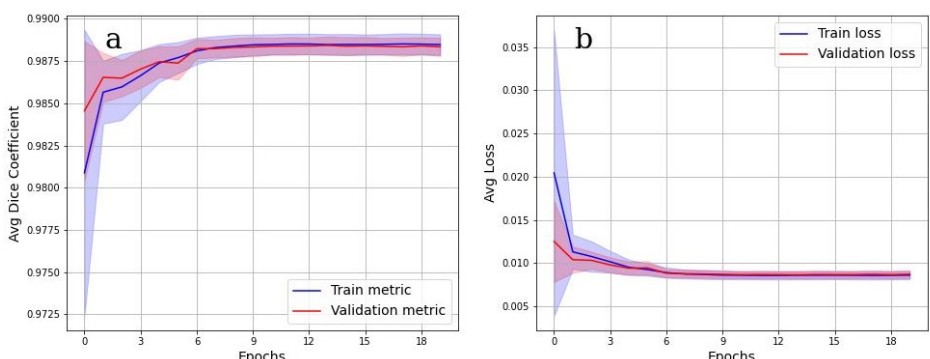

**Figure 7.** Binarization results. (**a**) Dice coefficient on training and validation datasets. Binarization quality increases as the coefficient approach 1; (**b**) loss curve on a training and validation dataset over 5 epochs.

The accuracy of multiclass segmentation (CNN-based labelling (segmentation) of the OB layers) was slightly less than the accuracy of binary segmentation (OB segmentation from the background). The curve corresponding to the validation dataset reaches a plateau by the sixtieth epoch. The accuracy on the training model began to increase faster than on the validation one, which indicates the beginning of retraining. The accuracy on the dataset has approximately reached a plateau. Given this, as well as a small dataset, further training cannot be considered as appropriate.

Cross-validation results in Figure 8a,b show, however, that the model was well trained. There was also no overfit since the accuracy and loss of the segmentation on the validation set were close to the accuracy and loss on the train set. An integral accuracy of 80% for all classes can be considered as acceptable. Cross-validation Dice value for the testing is 0.84.

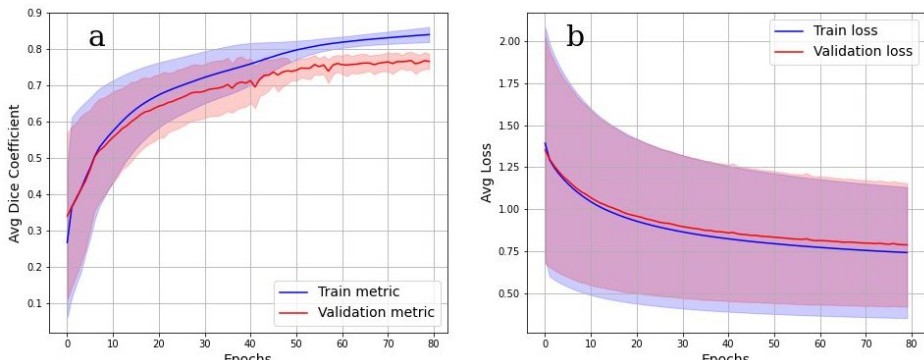

**Figure 8.** Multiclass OB segmentation. (**a**) Dice coefficient for training and validation datasets, (**b**) multiclass loss on training and validation datasets.

## 4. Discussion

In this work, XPCT imaging was used to visualize and segment 3D morphological structures in native unstained OB collected from a healthy non-demented donor. This research is a starting point for our studying of change in the morphology of the OB in relation to olfactory dysfunction. A wide variety of publications based on different methodologies and techniques was dedicated to the study of impaired olfaction and faster cognitive decline in neurodegenerative diseases [38,39]. However, the mechanism of the OB involvement in neurodegeneration is still unclear. In this framework, the XPCT technique, enabling non-invasive high-resolution 3D imaging of untreated brain tissue, serves as a bridge between two gold-standard imaging methods, viz. non-invasive 3D low-resolution MRI technique and high-resolution 2D histological imaging with invasive sample preparation.

Most of the knowledge about OB comes from animal research. Though the human and mammal OBs share similar morphology, the specific architecture of the human OB makes 3D image segmentation rather a challenging task. Both human and animals OBs are organized into laminations with layers representing specific neuron types, neuronal processes, and connections [40]. While in the mouse, the laminations in the OB are evident, in the human OB, the cellular and synaptic organization in layers is less strict. In addition, compared with rodents, the human OB often lacks circular organization of laminations and medial-lateral symmetry [41]. Moreover, OBs from older donors and peoples with impaired olfaction, in general, tended to exhibit more atypical or aberrant patterns of morphological organization. In this regard, they need a high resolution technique such as the XPCT, that permits it to accurate discern OB layers. In fact, CT and MRI provide morphometric information on OB, but these well-established techniques are limited by contrast agent requirements or low resolution of about 2 millimeters [42,43].

A conventional segmentation approach such as thresholding (see Supplementary materials) and manual segmentation of layers in the XPCT image of human OB is a difficult task due to a variety of concerns. In particular, one can face in the XPCT image a similar grayscale level between soft tissues, reconstruction artefacts, noise in image etc. At the same time, large population-based studies with extensive variability among the human brain require the processing of huge amounts of data. Therefore, the automatic segmentation of the OB structures starts to be an inevitable step. To solve the segmentation problem, we proposed to use a CNN-based algorithm. In our work, the CNN model was built and then pre-trained using the OB from one donor. In the future, our model should be adapted for a larger set of the OB samples accounting for significant anatomical variety OB morphology in young and elderly donors.

The proposed segmentation method is an important research tool for the comparative study of OBs, since it enables one to distinguish morphological layers in OBs and to investigate specific changes within a layer. However, there is still the need for further development of the algorithm to facilitate investigation of the OB neural and vascular

networks via a high-resolutions XPCT image of the OB. The last studies on the mechanisms of the OB degeneration in aging, as well as in Alzheimer's Disease (AD) and COVID-19 clearly demonstrate a need for a high-resolution sub-microstructural assessment of the OB tissue [44,45].

Thus, a structural study of the OB in olfactory dysfunction demonstrated a layer-specific neuronal loss in the glomerular layer [46], and degenerative changes in OB neurons has been found in AD [47]; the post-mortem high-resolution MRI of the brains affected by SARS-CoV-2 and histopathological examination demonstrated microvascular changes in OB [48]. Therefore, future studies on OB in a normal state and pathology will include investigation of the both gross- and micromorphology. Dynamic changes in OB structure will be a centre of interest as well. As a result, to provide an objective study of the OB degeneration in patients with olfactory disturbance, we intend in the future to develop fully automated segmentation of the micro-morphological structure such as different cell populations, vascular and capillary networks in each OB layer. The aim of our study was to understand whether the idea of automated segmentation of the OB layers within XPCT is feasible in terms of artificial intelligence. From this perspective, we have demonstrated a promising result. However, since the training, validation, and test image dataset were all from the same block of specimen, the model performance on independent datasets is yet to be studied. Potentially, the following problems can arise when using a limited data set: low robustness in relation to unique casesa and low stability if experimental conditions vary. Thus, our pre-trained model might benefit from including additional XPCT datasets in the pipeline, allowing the provided model to be tailored to a specific dataset. The code for inference of the trained model and additional information are available for free at [37].

## 5. Conclusions

In this paper, we present the results of the automated segmentation of the OB layers within XPCT images. The segmentation is a final output from the pipeline developed by us to assist the morphological examination of the OB.

The segmentation method is based on the U-Net convolutional neural network architecture. The developed pre-trained models allow us to sequentially solve two problems. The first one is the binarization of the gray scale image of the XPCT slice. The binarization split the image into the object (OB) and the background. The second one is a multi-class object segmentation. Each class corresponds to one layer of the OB.

We believe that the proposed automatic segmentation methods could provide new perspectives in the comparative study of OBs in olfaction impairment research of physiological brain aging, psychiatric and neurodegenerative diseases, and bacterial or viral infections.

**Supplementary Materials:** The following are available online at https://www.mdpi.com/article/10.3390/tomography8040156/s1.

**Author Contributions:** A.M. and A.K., software writing, data curation, and text writing; A.B. and I.B., experimental data acquisition, data curation, software validation, and text writing; I.B., XPCT image reconstruction and data analysis; O.J., sample preparation, experimental data acquisition, preparing training dataset, validation, and text writing; M.F., concepted and performed XPCT experiment and funding acquisition; A.C., methodology, conceptualization, and funding acquisition; M.C., validation, text writing, and funding acquisition; A.Y., validation and text writing; G.G., methodology and funding acquisition; F.W. and E.L., experimental support; V.A., formal analysis and conceptualization; S.S., methodology and project administration; D.N., methodology and project administration. All authors have read and agreed to the published version of the manuscript.

**Funding:** CNR-RFBR: CUP B86C17000460002 & Russian number 18-52-7819; RFBR: 18-29-26028; MIUR/CNR: CUP B83B17000010001; Regione Puglia: CUP B84I18000540002.

**Institutional Review Board Statement:** The study was carried out on autopsy material obtained from the collection 96 of Federal State Scientific Institution Research Institute of Human Morphology (Moscow, Russian Federation). All protocols were approved by the Ethical Committee of the Research Institute of Human Morphology of the Russian Academy of Medical Sciences (now FSSI Research

Institute of Human Morphology) (No. 6A of 19.10.2009). They are in correspondence with instructions of the Declaration of Helsinki, including points 7–10 for human material from 12.01.1996 with the last amendments from 19.12.2016.

**Data Availability Statement:** The code for inference of the trained model and additional information are available for free at https://github.com/ankhafizov/olfactory-bulb-segmentation, accessed on 8 June 2022.

**Acknowledgments:** The bilateral project CNR/RFBR (2018–2020)—accordo CNR-RFBR delle Relazioni Internazionali (CUP B86C17000460002 & Russian number 18-52-7819), the national project RFBR (number 18-29-26028)-Russian Federation. The FISR Project "Tecnopolo di nanotecnologia e fotonica per la medicina di precisione" (funded by MIUR/CNR, CUP B83B17000010001) and the TECNOMED project (funded by Regione Puglia, CUP B84I18000540002) are acknowledged for financial support. We also thank the Ministry of Science and Higher Education of Russian Federation within the State assignment FSRC «Crystallography and Photonics» RAS for financially supporting our work on tomography algorithms development.

**Conflicts of Interest:** The authors declare that they have no known competing financial interest or personal relationships that could have appeared to influence the work reported in this paper.

## Abbreviations

The following abbreviations are used in this manuscript:

| | |
|---|---|
| OB | Olfactory bulb |
| XPCT | X-ray phase-contrast tomographic |
| DL | Deep learning |
| CT | Computed tomography |
| MRI | Magnetic resonance imaging |
| AD | Alzheimer's disease |
| PD | Parkinson's disease |
| CNN | Convolutional neural network |
| ROI | Region of interest |
| FBP | Back Projection method |
| ReLU | Rectified Linear Unit |
| GL | Glomerular layer |
| EPL | External plexiform layer |
| MCL | Mitral cell layer |
| IPL | Internal plexiform layer |
| GL | Granule cell layer |
| AON | Anterior olfactory nucleus |
| MC | Mitral cell |

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
