# Peer review of "Deep Learning-Based Segmentation of Post-Mortem Human’s Olfactory Bulb Structures in X-ray Phase-Contrast Tomography"

_tomography, doi:10.3390/tomography8040156_

Round 1

Reviewer 1 Report

Unfortunately the paper is not suitable for publication because the machine learning methodology is too weak for the result to be believable to outsiders. After redoing several things, it may be suitable for publication, however.

The authors labeled 300 images and split them into training and ‘validation sets’. They should redo with the standard approach of training, val, and test sets. It is possible to overfit on a validation set based on when you stop the training and if multiple runs are done with the same validation set it is easy to pick the one with the best validation score during the course of research. Thus, splitting into train, val, and test is essential. Using only 30 images for testing is small. I suggest redoing with 50 test images and 20-30 for validation. This will leave less for training so your test dice may be lower, but the result will be more believable for readers. 

Ideally, the authors should use at least 4-fold cross validation - i.e. repeat the training 4 times with different randomized train-val-test splits and then report the average Dice across the results. I don’t think this is required here though since the tasks here appear visually to be quite easy, high performance is not surprising, and it appears the variance across folds should be low. 

Also it would be nice if the authors ran on some external (unlabeled) data and inspected the results to test the limits of their model. If they observed any failure modes on external data, they should report them. I am worried that their model is overfit (in particular the multiclass model, the foreground-background task is trivial). 

Consider including relative average area error in addition to Dice. It is defined as A_predicted - A_true / A_true , averaged over all images. The results may surprise you as this metric captures things differently than Dice.

The authors should say what batch size they used. They should clarify the stopping method (lines 180-181) since it isn’t easy to understand the way they explain it. To me the model looks overtrained, I would have stopped at 9 ~epochs for both models. How many layers were in the U-Net? What was the total number of trainable parameters in the model? Did you use dropout or a related method (drop block , drop connect) in addition to L2? If you based your architecture on model code that is online , please cite. Was the model in PyTorch or Tensorflow or some other framework? These details will help other researchers reproduce your work. 

The authors need to explain what the utility is here. The application is interesting, but what can be done with this model? What sorts of future scientific studies are enabled by this model? 

The authors should consider open sourcing their code and /or models for reproducibility reasons. 

The authors state: 

“Going deep into details, the stability of the segmentation is caused by 167 a relatively small number of tuning CNN weights and the existence of skip-connections, 168 e.g., informational channels between activation map in encoder and decoder. Both of these 169 features help U-Net to not overfit training data, e.g., avoid performing well on only the 170 training dataset, while failing on the validation dataset at the same time.” 

Honestly I am not sure any of this is true. Skip connections help preserve details but I am not sure if they help with robustness or overfitting. Please cite some references for these statements or remove. 

The paper could benefit from a thorough proofreading. There is a high density of minor errors, especially in the first 1-2 pages: 

In abstract: change “was” to “were” :  “Convolutional neural networks (CNN) was”, change “image” to “images” in the same sentence. 

“Deep Learning” should not be capitalized. 

Check the institution names. “Croc_code” doesn’t make much sense. 

Line 21: it highly -> ‘it is a highly” or “it’s a highly” 

Line 22: “in the segmentation” -> “for the segmentation”

Author Response

Comments and Suggestions for Authors

Unfortunately the paper is not suitable for publication because the machine learning methodology is too weak for the result to be believable to outsiders. After redoing several things, it may be suitable for publication, however.

 Thank you very much for your comments. We have tried to make clear several poits. 

The authors labeled 300 images and split them into training and ‘validation sets’. They should redo with the standard approach of training, val, and test sets. It is possible to overfit on a validation set based on when you stop the training and if multiple runs are done with the same validation set it is easy to pick the one with the best validation score during the course of research. Thus, splitting into train, val, and test is essential. Using only 30 images for testing is small. I suggest redoing with 50 test images and 20-30 for validation. This will leave less for training so your test dice may be lower, but the result will be more believable for readers. 

The validation set was not chosen randomly. The part of the layers of the bulb was chosen for the train and the other for the validation. Thus, the neural network is validated on a set of layers that differ markedly from the training examples. Therefore, the proposed validation sample is sufficient. Additionally, trained network was tested on unlabeled (external) data by specialist. There did not found an overfitting.

Ideally, the authors should use at least 4-fold cross validation - i.e. repeat the training 4 times with different randomized train-val-test splits and then report the average Dice across the results. I don’t think this is required here though since the tasks here appear visually to be quite easy, high performance is not surprising, and it appears the variance across folds should be low. 

We have done the cross-validation experiments during the investigation and the results obtained are presented in Figure 1.

Fig.1 Cross-validation average Dice

 Also it would be nice if the authors ran on some external (unlabeled) data and inspected the results to test the limits of their model. If they observed any failure modes on external data, they should report them. I am worried that their model is overfit (in particular the multiclass model, the foreground-background task is trivial). 

The main goal that we set for ourselves was not to build a model for segmenting any olfactory bulbs, but to make it easier for the expert to mark up the data. After the time-consuming labeling of several layers by an expert, the neural network performs the rest of the labeling, which is then verified by the expert. Unfortunately, it is not possible to test our model on other datasets. The fact is that the olfactory bulbs measured at synchrotron sources are a unique object data and we do not know of any other external dataset than the one we have. An effective pixel size is 1.28x1.28 μm2.

Consider including relative average area error in addition to Dice. It is defined as A_predicted - A_true / A_true , averaged over all images. The results may surprise you as this metric captures things differently than Dice.

The Dice metric is widely used to compare discrete images. Dice allows evaluation of a percentage classification error on image. The proposed metric showed the high efficiency of the neural network. For further development of neural network approach by researcher community, all source code and training data are published in an open github repository.

The results obtained with proposed metric are presented in Fig2. Dynamic of average value isgiven.

Figure 2. Results of cross-validation experiment with proposed metric.

The authors should say what batch size they used.

Batch size is 1.

They should clarify the stopping method (lines 180-181) since it isn’t easy to understand the way they explain it. To me the model looks overtrained, I would have stopped at 9 ~epochs for both models.

We have used the result of the model work with the best value of the loss function for 20 epochs on the validation set. In accordance with the Figure presented in the paper, the minimum on the validation set was reached at epoch 11.

How many layers were in the U-Net?

The architecture from the work [https://arxiv.org/pdf/1505.04597.pdf] was used. In total, the network has 23 convolutional layers.

What was the total number of trainable parameters in the model?

The model has approximately 17.3 million trainable parameters.

Did you use dropout or a related method (drop block , drop connect) in addition to L2?

No, we did not use. The current form of the loss function on the validation set looks reasonable.

If you based your architecture on model code that is online , please cite.

The reference on the corresponded paper was included in the list. The reference number is 26.

Was the model in PyTorch or Tensorflow or some other framework? These details will help other researchers reproduce your work. 

 The architecture of the Unet network was written using the Pytorch library, based on the open source implementation of Unet https://github.com/milesial/Pytorch-UNet.

The authors need to explain what the utility is here. The application is interesting, but what can be done with this model? What sorts of future scientific studies are enabled by this model? 

 We have added the explanations in the text (the lines from 24 to 30).

The authors should consider open sourcing their code and /or models for reproducibility reasons. 

We have added  https://github.com/ankhafizov/olfactory-bulb-segmentation.

The authors state: 

“Going deep into details, the stability of the segmentation is caused by 167 a relatively small number of tuning CNN weights and the existence of skip-connections, 168 e.g., informational channels between activation map in encoder and decoder. Both of these 169 features help U-Net to not overfit training data, e.g., avoid performing well on only the 170 training dataset, while failing on the validation dataset at the same time.” 

Honestly I am not sure any of this is true. Skip connections help preserve details but I am not sure if they help with robustness or overfitting. Please cite some references for these statements or remove. 

We have added:

Drozdzal, Michal, et al. "The importance of skip connections in biomedical image segmentation." Deep learning and data labeling for medical applications. Springer, Cham, 2016. 179-187.

The paper could benefit from a thorough proofreading. There is a high density of minor errors, especially in the first 1-2 pages: 

 Thank you very much for your comments. We have corrected.

Reviewer 2 Report

Dear Authors, 

The manuscript titled "Deep LeaRning-based segmentation of post-mortem human’s olfactory bulb structures in X-ray phase-contrast tomography" is well written and structured.

Up-to-date, there are many paper focusing on micro-CT for soft tissues that are able to obtain very high quality morphology (doi: 10.3390/app12104918). The Authors should underline the better quality of their XPCT techniques in comparison to other methodology.

On this regard, since the Authors aim to investigate the micro-structure of the olfactory bulb (that is noteworthy), there are no results indicating or showing this. 

Indeed, as reported in the main figures reported in the text, only different slices in 2d are present. It should be interesting that Authors show also the 3d images obtained.

Furthermore, it should be interesting if the Authors have data on altered olfactory bulb, in order to better understand if their tool effectively help in discriminating, and how, healthy and damaged olfactory bulbs.

Some typos are present starting from the title and the keywords (deep LEANING should be replaced with LEARNING). Please read carefully the manuscript again fixing the typos.

Finally, in the abbreviations title, I suggest the Authors to add all the abbreviations used in the manuscript such as, XPCT, DL, and so on.

Author Response

The manuscript titled "Deep LeaRning-based segmentation of post-mortem human’s olfactory bulb structures in X-ray phase-contrast tomography" is well written and structured.

Up-to-date, there are many paper focusing on micro-CT for soft tissues that are able to obtain very high quality morphology (doi: 10.3390/app12104918). The Authors should underline the better quality of their XPCT techniques in comparison to other methodology.

We have tried to make clear our position. We have added in the text lines 23-31, 53-56.

On this regard, since the Authors aim to investigate the micro-structure of the olfactory bulb (that is noteworthy), there are no results indicating or showing this. Indeed, as reported in the main figures reported in the text, only different slices in 2d are present. It should be interesting that Authors show also the 3d images obtained.

We have added Fig.6.

Furthermore, it should be interesting if the Authors have data on altered olfactory bulb, in order to better understand if their tool effectively help in discriminating, and how, healthy and damaged olfactory bulbs.

We believe, that we will have a chance to apply our approach as soon as the first measurements with these samples will be curry out.

Some typos are present starting from the title and the keywords (deep LEANING should be replaced with LEARNING). Please read carefully the manuscript again fixing the typos.

Thank you very much. We have done.

Finally, in the abbreviations title, I suggest the Authors to add all the abbreviations used in the manuscript such as, XPCT, DL, and so on.

Thank you very much. We have done.

Round 2

Reviewer 1 Report

The authors did not do a train-val-test split as I requested or cross validation.   

Furthermore the authors say: "The part of the layers of the bulb was chosen for the train and the other for the validation. Thus, the neural network is validated on a set of layers that differ markedly from the training examples"

--- This is very non-standard and somewhat questionable of a method, in my opinion.   

the authors should at least do a train-val-test with random sampling as is  the normal convention.

the authors also say "it is not possible to test our model on other datasets..."

This calls into question the utility of their model, there are not a lot of data where it can be used.

Author Response

Comments and Suggestions for Authors

We would like to thank the reviewer for the constructive suggestions and comments. We largely agree with the points raised and, according to comments from the reviewer, we provide some clarifications here. Below, we give our explanations, responding to the comments of the reviewer point-to-point.

  • The authors did not do a train-val-test split as I requested or cross validation.   

Furthermore the authors say: "The part of the layers of the bulb was chosen for the train and the other for the validation. Thus, the neural network is validated on a set of layers that differ markedly from the training examples"

This is very non-standard and somewhat questionable of a method, in my opinion.   

  • the authors should at least do a train-val-test with random sampling as is the normal convention.

We did a train-val-test experiments with random sampling. Train set contained 240 images, validation set contained 30 images and test set contained 30 images. The results are presented below. The paper part “CNN training results” was changed according to the results obtained.

Figure 1. Binary segmentation. Cross-validation results. Dice coefficient.

Figure 2. Binary segmentation. Cross-validation results. Loss value.

Figure 3. Binary prediction. Image was extracted from the test set.

Figure 4. Multiclass segmentation. Cross-validation results. Dice coefficient.

Figure 5. Multiclass segmentation. Cross-validation results. Loss value.

Figure 6. Multiclass prediction. Image was extracted from the test set.

  • the authors also say "it is not possible to test our model on other datasets..." This calls into question the utility of their model, there are not a lot of data where it can be used.

The main utility of the developed model for OB segmentation is to facilitates the work of biologists. Morphometric analysis using conventional 3D imaging tools such as MRI are of little use due to low spatial resolution. To our knowledge, our XPCT study is the first to show high-resolution 3D images from the whole OB to the cells. The indistinct boundaries of the OB layers pose a challenge to segregating (delineating) cell populations manually. In addition, manual segmentation is labor- and time consuming (each sample requires segmentation of several thousand slices). Here we present the first proof-of-concept results on the feasibility of DL-based modeling for segmenting OB morphological layers in high-resolution XPCT image. To begin, we used one sample to illustrate the possibility of a completely automatic solution to this problem.

In response to the referee's comments, the manuscript has been revised as follows. The following phrases have been added to Abstract:

“Human olfactory bulb (OB) has a laminar structure. The segregation of cell populations in the OB image poses a significant challenge because of indistinct boundaries of the layers.”

The text in Introduction was reorganized and revised to explain the results and utility of our model better. Also, we have included the following phrase:

“Because of indistinct layer boundaries, segregation of cell populations in the OB image is challenging issue.”

"This work represents a proof of concept for the development of a segmentaion tool for high resolution XPCT images."

Reviewer 2 Report

Dear Authors, 

The manuscript has been largely improved. However, just a last consideration and suggestion on my previous last point. 

In the abbreviation section, not all the abbreviated word used in the main text are present (such as CT, MRI, AD and so on). The Authors added just few. Please, add all the abbreviations or remove the subheading.

Author Response

Dear Authors,
The manuscript has been largely improved. However, just a last consideration and suggestion on my previous last point.
In the abbreviation section, not all the abbreviated word used in the main text are present (such as CT, MRI, AD and so on). The Authors added just few. Please, add all the abbreviations or remove the subheading.
We thank the reviewer for their comments. We have corrected the text as follow their suggestions.
We have supplemented the abbreviation section.
